# Mesothelioma in Agriculture in Lombardy, Italy: An Unrecognized Risk

**DOI:** 10.3390/ijerph18010358

**Published:** 2021-01-05

**Authors:** Carolina Mensi, Barbara Dallari, Marco Polonioli, Luciano Riboldi, Dario Consonni, Angela Cecilia Pesatori

**Affiliations:** 1Occupational Health Unit, Fondazione IRCCS Ca’ Granda Ospedale Maggiore Policlinico, 20122 Milan, Italy; barbara.dallari@policlinico.mi.it (B.D.); luciano.riboldi@unimi.it (L.R.); dario.consonni@unimi.it (D.C.); angela.pesatori@unimi.it (A.C.P.); 2Post Graduation School of Occupational Medicine, Università degli Studi di Milano, 20122 Milan, Italy; marco.polonioli@virgilio.it; 3Department of Clinical and Community Science, Università degli Studi di Milano, 20122 Milan, Italy

**Keywords:** malignant mesothelioma, asbestos exposure, agriculture, occupational cancers

## Abstract

Cohort studies showed consistently low risks for malignant mesothelioma (MM) among agricultural workers, however the investigated exposures did not include asbestos. Our aim is to describe sources of asbestos exposure of MM in agriculture. Twenty-six MM cases in agricultural or seed trades workers were identified through the MM registry of the Lombardy region, Italy in 2000–2016. Asbestos exposures were investigated through a standardized questionnaire. The most frequent exposure circumstances were recycled jute bags previously containing asbestos (11 cases) and maintenance and repair of asbestos roofs (12 subjects). Three subjects performed maintenance and repair of tractor asbestos brakes and two used asbestos filters for wine production. Our data suggest asbestos exposure opportunities in the agricultural setting, underlining the need to look for this exposure in subjects affected with mesothelioma.

## 1. Introduction

Malignant mesothelioma (MM) is a rare cancer that arises from serous tissues of pleura, peritoneum, pericardium and tunica vaginalis of testis and is closely related to asbestos exposure. For this reason it is considered a ‘sentinel event’ that strongly suggests the presence of asbestos exposure. This neoplasm has a very poor prognosis, is refractory to therapies and appears after a long time since first exposure (latency period) of asbestos. Epidemiological surveillance of mesothelioma is realized by a specialized cancer registry, that in Italy has been compulsory since 2000 [1]. The availability of an archive of MM cases and their asbestos exposure histories is useful not only to confirm the numerous known uses of this mineral, but also to describe unknown exposure circumstances [2]. 

Several studies have examined cancer risk in agricultural populations or farmer workers. The largest cohort studies are the Agricultural Health Study (AHS) in USA [3], the AGRICAN cohort study in France [4], the Finnish farmer cohort [5] and the Canadian census health and environment cohort (CanCHEC) [6]. Compared with the general population, updated mortality and incidence findings have been published [3,4,6], showing a quite consistent pattern of reduced risk for all cancer and smoking related cancer (lung, larynx, bladder). Increased risks for specific cancer sites (prostate, brain, lip, skin and lymphatic and hematopoietic neoplasms) have been reported but definite conclusions are still lacking. In 2010, a consortium of agricultural cohort studies from 9 countries (AGRICOH) has been established to promote collaboration and data pooling to allow research on rare outcomes and specific exposure-disease associations [7].

Pesticides have been the most widely examined exposures along with ultraviolet radiation, viral and bacterial exposures related to livestock and poultry, dust, and diesel exhaust [8,9]. However, asbestos is not currently considered as an occupational risk factor among farmers.

As shown in Table 1, the most updated findings in the cohorts mentioned above showed consistently decreased risks for malignant mesothelioma in agriculture compared with other sectors. However, no study focused specifically on the relationship between asbestos exposure in agricultural settings and mesothelioma occurrence. Therefore the aim of this manuscript is to describe sources of asbestos exposure in agricultural workers diagnosed with malignant mesothelioma in the period between 2000 and 2016 in the Lombardy region (Italy), based on the systematic collection of mesothelioma cases and their asbestos exposure through a specific registry. 

## 2. Material and Methods

MM cases were identified through the MM Registry of the Lombardy Region (RLR). The RLR is a population based registry, that since 2000 has collected all incident cases of MM (including pleura, peritoneum, pericardium and tunica vaginalis testis) diagnosed in subjects residing in the Lombardy region (currently about 10 million inhabitants). Pathology, pneumology, surgery and oncology regional departments actively report to the RLR each suspected case. Completeness of reporting is assured by periodic linkage with databases of pathology departments, hospital discharge records, mortality registries and occupational disease compensation from INAIL (Italian Working Compensation Authority) [10]. The final diagnosis of each case is established through a panel discussion based on all available clinical, pathology and imaging information. According to the guidelines of the Italian Mesothelioma Registry [11] cases are classified as ‘definite MM’ (histological diagnosis and imaging), ‘probable MM’ (usually, cytology suggesting MM plus imaging); ‘possible MM’ (positive imaging). Morphology is defined and coded according to WHO histological classification [12]. 

Asbestos exposure is ascertained through a standardized questionnaire administered to the patient, or to the next of kin, by trained interviewers. The questionnaire is designed to obtain a complete occupational history including the industrial sector, plant, job, and specific tasks performed. Besides complete residential history, lifestyle habits and home related activities (small repair works, thermal insulation, building or machine maintenance, ironing on asbestos-coated ironing boards), are investigated for each patient. Information regarding the jobs performed by each cohabitant are also collected. On the basis of this information, the lifetime asbestos exposure is classified as ‘occupational’ (definite, probable, possible), ‘para-occupational’ (exposure through the cohabitants), ‘home related’ or ‘environmental’. Occupational exposure is classified as ‘definite’ for subjects who have carried out a work activity involving the use of asbestos or asbestos-containing materials; ‘probable’ when the subject worked in an industry or workplace where asbestos was used but personal exposure cannot be documented; ‘possible’ for subjects who have worked in an economic sector in which the presence/use of asbestos has generally been found, but there is no information on the use or otherwise of asbestos by them.

The vital status of each case is ascertained through local health units and survival time is calculated based on the date of diagnosis and date of death (or last follow-up for alive subjects).

First, we selected all incident MM cases in the period 2000–2016 who had been employed in the agricultural sector or in seed trades; then, we described the cases with exposure to asbestos exclusively in these working sectors. 

Ethics Statements: Malignant mesothelioma reporting to registry is compulsory by law (277/1991 and 81/2008), therefore ethics approval is not required.

## 3. Results

### 3.1. Characteristics of Study Population

In the period 2000–2016 the RLR collected 6237 MM cases (4054 males and 2183 females). Among these, 36 had at least one work period with recognized asbestos exposure in the agricultural sector or in seed trades. We selected the 26 cases who had been exposed to asbestos exclusively in the agricultural sector or in seed trades (i.e., we excluded subjects with occupational asbestos exposure also in other working sectors or in non-occupational settings). 

The main characteristics of these MM are described in Table 2: most were males, 25 had pleural mesothelioma and age at diagnosis ranged from 56.8 to 85.7 years. The most frequent histology was epithelial mesothelioma. Seven males also had pleural plaques in addition to MM. Diagnosis was classified as definite in 20 subjects.

Information on asbestos exposure was collected directly from the patient for most cases (80.8%) and occupational asbestos exposure was classified as definite in 69.2% of cases. The latency time (time since first exposure to asbestos) was 49.4 and 58.8 years in males and females, respectively. Median length of exposure was 27.6 and 38.6 years in males and females, respectively.

All subjects died after a median survival time of 12.3 months for males and 10.6 months for females. Survival was longer for the 16 subjects with epithelioid MM (16.3 months, 95% CI: 6.2–29.7), compared with the four subjects with biphasic MM (5.7 months) and those without morphologic diagnosis (4.0 months).

### 3.2. Circumstances of Asbestos Exposure

Circumstances of asbestos exposure are described in Table 3. Those most reported were the use of recycled jute bags that previously contained asbestos mineral and then were used for vegetables, cereals, seeds and feed packaging (11 cases) besides maintenance and repair of asbestos roofs in agricultural buildings (12 subjects). A few subjects (three) had exposure due to maintenance and repair of tractors with asbestos brakes or frictions and gasket and to the use of asbestos filters for wine production (two subjects). Seven cases had bilateral pleural plaques, a well-known marker of asbestos exposure: three used recycled jute bags, three have been involved in roof installation and repair, one in repair and maintenance activities of tractors and other agricultural machinery for his farm and other nearby farms. For most cases (19, 73%) exposure began in the years on or prior to 1970.

Nineteen cases (82.3%) filled-in compensation claims to the Italian Working Compensation Authority (INAIL), but only five were compensated (most were informal workers).

## 4. Discussion

Asbestos ban in Italy occurred in 1992 (Law nr 257, 27 March 1992) and prohibited extraction, import, export, marketing and production of asbestos or asbestos containing products. Despite the ban, the presence of asbestos containing materials in numerous worksites, machines and equipment may still constitute a source of exposure in industrial and also rural environments.

This brief description of MM in the agricultural sector confirms the existence of asbestos hazard in this sector. Several exposure sources exist in agriculture. The setting up of large canopies or rural premises covered with asbestos cement slabs was a low-cost solution for the shelter of animals, land products and vehicles in the 50s and 70s. Often their construction, maintenance and repair were not carried out by specialized construction companies but by farmers themselves also in most recent years [13]. Patients reported that removed slabs were reused to build shelters for tools or small houses for animals (such as chicken coops or rabbit hutches). Moreover, deteriorated slabs were not eliminated: they were broken and placed on the beaten earth to create paved paths, or to delimit parts of the garden or vegetable garden. 

The likelihood of exposure to asbestos dust also occurred in three cases performing repairs and maintenance on tractor brake pads and linings, which are known to contain asbestos for durability and fire proofing. In the agricultural sector it is customary to carry self-made maintenance activities without implementing adequate safety and health protection rules. Moreover, in our case series the year of first exposure to asbestos was well before the 1980s, a period in which, in Italy, asbestos containing materials were used without precautions.

Use of recycled jute bags as source of asbestos exposure has been documented in numerous Italian reports [14,15,16]. Until the 1970s asbestos was transported in sacks of textile fiber, mainly jute, which were then treated to be used for harvesting and trading of agricultural products such as cereals, olives and carobs. Our data confirmed this activity as a frequent occasion of exposure in farmers. 

Finally, another well-known source of exposure to asbestos for farmers was that associated with use of powdered asbestos to package filters for the treatment of wine as confirmed in two of our cases (Case N. 12 and 19 of Table 3) [17,18].

In all circumstances patients referred that personal protection devices were not used due to lack of awareness of asbestos presence and/or of its health effects. 

In this study, for the first time to our knowledge, 26 cases of MM were described that had exposure to asbestos exclusively in the agricultural sector. Notably, 27% of them have had also pleural plaques.

Identification of asbestos exposure was possible thanks to the use of a standardized questionnaire used by the National mesothelioma registry and to interviews performed by trained interviewers. A simple job-exposure matrix approach would not have allowed us to identify the particular tasks involving such exposure. Our study has some limitations. First, some affected subjects had died or were too ill to respond, and therefore the interview had to be administered to the next-of-kin, who may not be fully aware of circumstances of exposure. Second, although interviewers are trained, we cannot guarantee optimal interview quality in all circumstances. Third, assessment of asbestos exposure was only qualitative. Finally, no information was available on the type of asbestos fibers.

## 5. Conclusions

Our data underline the existence of multiple sources of exposure to asbestos in the agricultural setting and the need to thoroughly look for them in subjects affected with mesothelioma, particularly for subjects who worked before the 1980s when asbestos exposure opportunities were more frequent and overlooked. In absence of our registry, mesothelioma in this series of patients would not have been considered an occupational disease and therefore none of them would have been compensated.

Lastly, it is important to carry out inspections and censuses on the state of conservation of asbestos containing manufacts still present on farms and to inform workers about this hazard by prescribing that any maintenance works be carried out exclusively by specialized personnel.

## Figures and Tables

**Table 1 ijerph-18-00358-t001:** Relative risk of mesothelioma in agricultural cohort studies.

Study (Reference)	Period of Follow-Up	Observed Incident Cases	Relative Risk	95% Confidence Interval	Reference
Agricultural Health Study	1995–2015	21	0.94	0.61–1.44	Lubin JH, et al. [3]
Finnish farmers	1995–2005	6	0.29	0.11–0.62	Laakkonen A, et al. [5]
AGRICAN cohort	2005–2011	17	0.36	0.21–0.58	Lemarchand C, et al. [4]
CanCHEC	1992–2010	20	0.57	0.36–0.90	Kachuri L, et al. [6]

**Table 2 ijerph-18-00358-t002:** Characteristics of mesothelioma cases in agriculture, Lombardy Mesothelioma Registry, 2000–2016.

	Men		Women	
Characteristic	N	%	N	%
Total	21	100	5	100
Age (years), mean (min-max)	70.0	(56.8–83.9)	74.6	(65.4–85.7)
Site				
Pleura	20	95.2	5	100
Peritoneum	1	4.8	0	0.0
Diagnosis				
Definite	16	76.2	4	80.0
Probable	2	9.5	0	0.0
Possible	3	14.3	1	20.0
Morphology (ICD-O code)*				
Epithelial (90523)	15	71.4	1	20.0
Fibrous (90513)	1	4.8	0	0.0
Biphasic (90533)	2	9.5	2	40.0
MM NOS * (90503)	0	0.0	1	20.0
Not available	3	14.3	1	20.0
Interview				
Patient	16	76.2	5	100
Next of kin	5	23.8	0	0.0
Pleural plaques				
Yes	7	33.3	0	0.0
Occupational asbestos exposure				
Definite	16	76.2	2	40.0
Possible	5	23.8	3	60.0
Length of exposure (years), median (min-max)	23.0	(4–52)	40.0	(15–51)
Time since first exposure (years), median (min-max)	51.0	(33–65)	60.0	(54–63)
Survival time (months), median (min-max)	12.3	(0.7–76.6)	10.6	(5.7–27.6)

* ICD-O: International Classification of Disease-Oncology.

**Table 3 ijerph-18-00358-t003:** Circumstances of exposure to asbestos for mesothelioma cases in agriculture, Lombardy Mesothelioma Registry, 2000–2016.

N, Gender	Period of Exposure	Exposure Classification	Pleural Plaques	Circumstances of Exposure
1, F	1946–1985	Definite	NO	Shelling and grinding of corn and other grains stored in jute bags. Before using, bags were beaten to remove dust and residues of previous materials they had contained. Use of about 20–30 thousand bags/year
2, F	1962–2010	Definite	NO	Farmer and cattle breeding. About 500 square meters of asbestos roofs were present in the rural setting. It happened that due to the weather the slabs broke and fell to the ground. They were reused to create a pavement to walk and to make borders of vegetable gardens
3, M	1959–2001	Possible	NO	Breeder of cattle, pigs and horses. Frequent use of recycled jute bags for feed
4, M	1950–1970	Definite	YES	Frequent use of recycled jute bags previously used to pack asbestos
5, M	1953–1996	Definite	NO	Chicken farmer. All rural building maintenance activities including asbestos roofing was done by the worker
6, M	1943–1957	Definite	YES	Frequent use of recycled jute bags previously used to pack asbestos
7, F	1937–1952	Possible	NO	Frequent use of recycled jute bags
8, M	1948–2000	Definite	YES	Repair and maintenance activities of tractors and other agricultural machinery for his farm and other nearby farms
9, M	1965–1999	Possible	NO	Chicken farmer. He performed annual boiler maintenance and cleaning
10, M	1957–2008	Definite	NO	Farmer tractor driver. He took care of the tractor maintenance himself. Between 1960–1970 he installed asbestos roof in rural buildings
11, F	1940–1980	Possible	NO	Farming of chickens and turkeys in a family run farm with use of recycled jute bags for feed
12, M	1972–1993	Definite	NO	Farmer tractor driver who take care of tractor maintenance. Use of asbestos filters for wine production. In 1985 he laid an asbestos roof
13, M *	1957–1976	Possible	NO	Use of recycled jute bags
14, M	1956–1970	Possible	NO	Use of recycled jute bags
15, M	1972–1995	Definite	NO	Farmer tractor driver. He also repaired the asbestos roof by cutting the slabs before placing them
16, M	1960–1968	Definite	YES	He collaborated with his father in the installation of asbestos roofs for a total of 1200 square meters (20 sheds for breeding 30 thousand chickens)
17, M	1970–1975	Definite	NO	He built asbestos roofs for sheltering agricultural tools and other materials
18, M	1950–1996	Definite	YES	Frequent use of recycled jute bags. In 1950 he built an asbestos roof for chicken coop and garage
19, M	1971–1975	Definite	NO	Winegrower. Use of asbestos filters in the wine production process. The worker referred the use of about 1000 kg of asbestos/year in the process of wine filtering. No use of personal protective equipment
20, F	1945–1996	Possible	NO	Use of recycled jute bags that she banged before using and sometimes even repaired
21, M	1965–2004	Definite	NO	Game breeder. He frequently repaired asbestos roofs of warehouses. Asbestos panels were also present to separate bird cages
22, M	1980–1993	Definite	NO	Mainly rice farmer. In the 80s and 90s he helped the specialist asbestos removing company to remove and replace asbestos roofs
23, M	1961–2000	Definite	NO	Farmer. He was responsible of the maintenance of all tractors including brakes and clutches replacement. In 1980 and 1982 he installed asbestos roofs in rural buildings
24, M	1977–1987	Definite	YES	Farmer. He repaired asbestos roofs of stables
25, M	1949–1996	Possible	NO	Sale of cereals and fertilizers that he bagged in recycled jute bags
26, M	1963–2000	Definite	YES	Farmer. He built asbestos roofs for stables (about 700 square meters)

* Peritoneal mesothelioma.

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
