# Peer review of "Mesothelioma in Agriculture in Lombardy, Italy: An Unrecognized Risk"

_ijerph, 2021, doi:10.3390/ijerph18010358_

Round 1

Reviewer 1 Report

The manuscript concerns an important issue which is the occurrence of malignant mesothelioma among agricultural workers in the Lombardy region. The Authors described sources of asbestos exposure (e.g. use of recycled jute bags previously used to pack asbestos, use of asbestos filters in the wine production process, repairing of asbestos roofs) which is the primary risk factor for malignant mesothelioma.

The aim of the research is clearly defined. The introduction is written properly and provides sufficient background on the topic. The methods and results to a large extent are described in a proper way.

The discussion is adequate but in my opinion, the novelty and importance of the Authors’ results should be emphasized even more strongly. Moreover, the comparison of conclusions resulting from the research with previous state of knowledge should be improved however, the report can also be published in its present form.

Overall, the manuscript contains findings that can help relate the occurence of mesothelioma among agricultural workers to different sources of asbestos exposure.

Author Response

Reviewer 1.

Comments and Suggestions for Authors

The manuscript concerns an important issue which is the occurrence of malignant mesothelioma among agricultural workers in the Lombardy region. The Authors described sources of asbestos exposure (e.g. use of recycled jute bags previously used to pack asbestos, use of asbestos filters in the wine production process, repairing of asbestos roofs) which is the primary risk factor for malignant mesothelioma.

The aim of the research is clearly defined. The introduction is written properly and provides sufficient background on the topic. The methods and results to a large extent are described in a proper way.

The discussion is adequate but in my opinion, the novelty and importance of the Authors’ results should be emphasized even more strongly. Moreover, the comparison of conclusions resulting from the research with previous state of knowledge should be improved however, the report can also be published in its present form.

Overall, the manuscript contains findings that can help relate the occurence of mesothelioma among agricultural workers to different sources of asbestos exposure.

We feel a bit uncomfortable in further emphasizing our results. We think we had sufficiently highlighted the merits of this study.

Reviewer 2 Report

A word such as 'therefore' is suggested to  add at the begening of line 49, and then, all this paragraph is moved to the end of line 48.

There are too many paragraphs in the part of 'discussion'. I suggest mereging some paragraph together.

Author Response

Reviewer 2.

Comments and Suggestions for Authors

A word such as 'therefore' is suggested to  add at the begening of line 49, and then, all this paragraph is moved to the end of line 48.

Done.

There are too many paragraphs in the part of 'discussion'. I suggest mereging some paragraph together.

Done.

Reviewer 3 Report

To help improve the quality of the paper and increase potential readers interest, the follwoing section will require revising; 

i. Section 3.1 Chracteristics of study population: it is not clear the number of partcipants that is been reported here.  There was mention of 36 considered before excluding the 10. it will be good to provide clear exclusion criteria applied in the material and methods section. 

ii. it will be better to present data in closed bracket (21/26) as parcentage instead

iii. Considering that data were sourced from next of kin in some instance, i expect to read at some point limitation to this approach but all through  the report this was not stated nor any limitation to the method adopted in general.  

Author Response

Reviewer 3.

Comments and Suggestions for Authors

To help improve the quality of the paper and increase potential readers interest, the follwoing section will require revising; 

  1. Section 3.1 Chracteristics of study population: it is not clear the number of partcipants that is been reported here.  There was mention of 36 considered before excluding the 10. it will be good to provide clear exclusion criteria applied in the material and methods section. 
  2. The two exclusion criteria for 10 subjects are shown in round brackets.
  3. it will be better to present data in closed bracket (21/26) as parcentage instead
  4. Done.

iii. Considering that data were sourced from next of kin in some instance, i expect to read at some point limitation to this approach but all through  the report this was not stated nor any limitation to the method adopted in general.  

We describe limitations in the Discussion section (lines 161-165).

This manuscript is a resubmission of an earlier submission. The following is a list of the peer review reports and author responses from that submission.

Round 1

Reviewer 1 Report

  1. The manuscript descripts the situation in Lombardy Region, Italy, then the title should be changed, for example, Brief Report Mesothelioma in agriculture in Lombardy Region: an unrecognized risk;
  2. Line 33-34: references cited in the whole sentence should be listed;
  3. Line 39: the whole sentence is suggested to connect to the former paragraph, put the word of ‘therefore’ (an example) at the begins, and change the word ‘report’ to ‘paper’;
  4. Table 1: the refence is suggested to be place in a separate line (the last line);
  5. Line 43: change to “Material and Method”;
  6. For the whole part of 'resulted' (“Results and Analysis” is more suitable) from the Line 75 to Line 111, subtitles are suggested to be used to understand the meaning clearly, and adding more analysis is appreciated;
  7. Line 131-133: two cases cited in this sentence, which was from table 3, should be marked/line out;
  8. For the paragraph of 'Discussion', the order is suggested changing: line 73-74 is move to the beginning and followed by 72-73;
  9. Move Line 113-114 to the beginning of Line 119, and move Line 135-135 to the part of ‘method’;
  10. It is suggested to discuss the potential source/s of exposure to asbestos for farmers, and to add more useful and constructive discussion related to MM and agriculture in the part of ‘Discussion’.

Reviewer 2 Report

The manuscript concerns an important issue which is the occurrence of malignant mesothelioma among agricultural workers in the Lombardy region. The Authors described sources of asbestos exposure which is the primary risk factor for malignant mesothelioma. Malignant mesothelioma is a rare but aggressive (rapidly growing) malignant tumor of the mesothelium. Moreover, due to non-characteristic symptoms, it is diagnosed late and the treatment quite often does not bring satisfactory results (the mesothelioma death rate is relatively high). Therefore, I think that the submitted report would be suitable for publication in the International Journal of Environmental Research and Public Health. However, there are few minor points which need to be clarified or improved before the publication:

  1. In the Introduction, there is a lack of information about malignant mesothelioma, e.g. why is interesting, dangerous, etc. Why did the Authors become interested in this cancer?
  2. Please improve the Discussion. Some of the sources of asbestos exposure in agriculture are quite obvious and well known so it may seem that the report does not bring any new knowledge. Therefore, the Authors should focus on better describing the novelty of the presented results. Comparison of conclusions resulting from the research with previous state of knowledge should be added.
  3. Insufficient information why the Authors’ research is important. Please provide more information about the importance of obtained results.

Overall, the manuscript shows some interesting results, but on the other hand, some of the findings are quite obvious and well known. I recommend the manuscript for publication after the above suggestions are addressed.

Reviewer 3 Report

This paper is inconclusive because it suffers from the following problems:

1. The number of mesothelioma studies is very low and there are no controls.  How many other people with the same type of employment had the same type of exposure?  Without that information we do not know if the incidence of mesothelioma is above background

2. The evidence of exposure is circumstaintal at best.  FOr instance "maintainace of asbestos roofs", how many years how many roofs? Obviously there is a huge difference if someone does that on a regular basis for years or as summer job.  Where those jobs done under proper controlled regulatory rules -which means no exposure, or where they done without any pracautions against current laws?  and so on

3. There is no hard science to demonstrate that asbestos exposure actually occurred: radiological imaging suporting exposure, lugn content analyses or at least an iron stain of a lung biopsy

4. Exposure in the agricultural sector can be related to asbestos or other carcinogenic fibers present in the terrain, so where did these people work and is there any asbestos in the areas where they work

5. A large number of roofs, wood stoves, iron pads, gloves, sewage and water collectors, etc. etc. were made of asbestos in italy especially in the North of Italy where Eternit the largest asbestos producer was present.  These people may likely have been exposed at home if they had a wood=stove isolated with eternit panels for example, and their agricultural occupation may have had nothing to do with the actual exposure.

In short this paper represent an hypothesis based on circumstantial evidence.